# Ribonucleotide and R-Loop Damage in Plastid DNA and Mitochondrial DNA during Maize Development

**DOI:** 10.3390/plants12173161

**Published:** 2023-09-02

**Authors:** Diwaker Tripathi, Delene J. Oldenburg, Arnold J. Bendich

**Affiliations:** Department of Biology, University of Washington, Seattle, WA 98195, USA; tripad@uw.edu (D.T.); delene@uw.edu (D.J.O.)

**Keywords:** DNA damage, DNA repair, glycation, mtDNA, ptDNA, RNase H, ROS

## Abstract

Although the temporary presence of ribonucleotides in DNA is normal, their persistence represents a form of DNA damage. Here, we assess such damage and damage defense to DNA in plastids and mitochondria of maize. Shoot development proceeds from meristematic, non-pigmented cells containing proplastids and promitochondria at the leaf base to non-dividing green cells in the leaf blade containing mature organelles. The organellar DNAs (orgDNAs) become fragmented during this transition. Previously, orgDNA damage and damage defense of two types, oxidative and glycation, was described in maize, and now a third type, ribonucleotide damage, is reported. We hypothesized that ribonucleotide damage changes during leaf development and could contribute to the demise of orgDNAs. The levels of ribonucleotides and R-loops in orgDNAs and of RNase H proteins in organelles were measured throughout leaf development and in leaves grown in light and dark conditions. The data reveal that ribonucleotide damage to orgDNAs increased by about 2- to 5-fold during normal maize development from basal meristem to green leaf and when leaves were grown in normal light conditions compared to in the dark. During this developmental transition, the levels of the major agent of defense, RNase H, declined. The decline in organellar genome integrity during maize development may be attributed to oxidative, glycation, and ribonucleotide damages that are not repaired.

## 1. Introduction

In order to properly contribute to cellular metabolism and heredity, DNA damage must be recognized and repaired. This objective is achieved in the nucleus, where there are only two copies of the genome in diploid cells and only one copy in haploid gametes. In the mitochondria and plastids, however, there are many copies of these organellar genomes, most of which in leaf cells carry unrepaired lesions of various types and appear to be defective copies [1,2]. 

All organisms carry information in DNA using deoxyribonucleotides and RNA using ribonucleotides. Ribonucleotides can also be found in DNA, where their temporary presence is part of normal metabolism. If not replaced by deoxyribonucleotides, however, persistent ribonucleotides can be detrimental to the cell and represent “DNA damage” [3,4,5]. 

There are three major processes whereby ribonucleotides may be introduced into DNA. 

(1)During DNA replication, DNA polymerases must distinguish between deoxyribose and ribose in the sugar moiety, as well as choose the correct complimentary base [6]. Misincorporation of ribonucleotides may occur, however, because of the inadequate ability of DNA polymerases to distinguish between these two sugars. The outcome of this misincorporation is typically a single ribonucleotide. Misincorporation rates can be fairly high, even in healthy cells [7,8]. Ribonucleoside monophosphate has been reported as the most common aberrant nucleotide found in DNA: one per 10,000 to 100,000 nucleotides [7,8,9,10,11]. Genome instability may be caused by the reactivity of the 2′-OH in the sugar portion of a ribonucleotide, which increases its susceptibility to strand cleavage [7].(2)For lagging-strand DNA synthesis, a short stretch of ribonucleotides (an RNA primer) anneals to the template strand and primes replication [12]. As DNA synthesis proceeds, the RNA primer is removed, the resulting gap is filled with deoxynucleotides, and the nascent strand is sealed by DNA ligase. If the RNA primer is not removed, however, it may be deleterious to the organism.(3)Another type of RNA-DNA structure may be generated during DNA replication, repair, and transcription: an R-loop. R-loops are three-stranded nucleic acid structures containing a displaced DNA strand and an RNA-DNA hybrid (RNA base-paired with its complementary DNA) that influence many biological processes [13,14,15]. In yeast, mammals, and plants, R-loops can occupy approximately 5 to 10% of the nuclear genome [16,17,18], suggesting that they are typically tolerated. Nevertheless, persistent R-loops may lead to DNA replication fork stalling, genome instability [19], DNA damage [20], chromosomal DNA rearrangements, recombination [21], and transcription–replication conflict [22]. The effects of persistent R-loops may result in human diseases, including cancer, neurodegeneration, and inflammatory diseases [15,23,24]. In summary, R-loops can affect genome stability.

There are two principal defense mechanisms to remove ribonucleotides in DNA and counteract their damaging effects. The primary components in these multi-protein repair systems are RNase H1 and RNase H2 [5,25]. Repair is initiated by the cleavage of the covalent bond between the ribo- and deoxyribonucleotides. Other components remove and replace the ribonucleotide(s) with deoxy forms or allow the displaced ssDNA to pair with its complement. In either case, the normal dsDNA segment is reconstituted. RNase H1 is a single-subunit protein and requires a stretch of at least four consecutive ribonucleotides for recognition and activity [4,26]. RNase H1 can remove R-loops adjacent to mitochondrial DNA (mtDNA) replication origins in mammals [27]. In contrast, the RNase H2 holoenzyme comprises three protein subunits and can recognize and hydrolyze a single ribonucleotide misincorporated during DNA replication [4,28,29,30]. RNase H2 can also recognize and repair a string of multiple ribonucleotide monophosphates (rNMPs) within RNA-DNA hybrid regions. Sequence comparisons have shown similarities among the RNase H proteins associated with plastids and mitochondria [4,11,31]. 

In the model plant, *Arabidopsis thaliana*, three RNase H1 proteins have been identified and are localized to the nucleus, mitochondria, and chloroplasts (respectively, AtRNH1A, AtRNH1B, and AtRNH1C) [28]. In plastids, other proteins, including DNA gyrase, Whirly, RecA, and RHOM1, have been associated with RNase H1 and are involved in restricting R-loops and maintenance of plastid genome integrity [28,32,33]. The Arabidopsis RNase H2 holoenzyme comprises three subunits (A, B, and C), with the A subunit showing sequence similarity to the *TRD1* genes in maize and rice; RNase H2 was shown to suppress nuclear genome instability by removing ribonucleotides and reducing the rate of homologous recombination [11]. To our knowledge, very little, if anything, is known about RNase H proteins in organelles of other plants, including maize.

In maize, organellar DNA (orgDNA) exhibited structural changes during leaf development and in response to light, whereby pristine multi-genomic molecules were degraded to sub-genomic fragments. This degradation was associated with orgDNA damage from oxidative and glycation reactions, as well as inadequate damage defense and DNA repair [34,35]. The shifting patterns of damage and damage defense were tightly coordinated during development [2]. Here, we report on a third type of orgDNA damage due to ribonucleotides, R-loops, and changes in damage defense from RNase H proteins during leaf development and light/dark growth conditions. The data show that this third type of damage defense follows a similar developmental pattern as oxidative and glycation damage defense. Furthermore, there do not appear to be any detrimental effects on the maize seedling under normal growth conditions due to the decline in high-integrity orgDNAs in the leaf. The results may, however, be relevant to the genetic engineering of maize because the paucity of undamaged orgDNA in leaf tissue may impede plant regeneration and plastid transformation. 

## 2. Materials and Methods

### 2.1. Plant Tissue and Growth Conditions

Maize [*Zea mays* (L.), inbred line B73] seeds were soaked overnight and sown in Sunshine soil Mix #4 and vermiculite (1:1 ratio) (Amazon Inc., Seattle, WA, USA). The seedlings were grown for 12 days with a 16 h light/8 h dark photoperiod (light-grown) or in continuous dark for 12 days (dark-grown) in a temperature-controlled room with light intensity ∼500 μmol s^−1^ m^−2^ photosynthetic photon flux density. Seedlings were washed with 0.5% sarkosyl (Sigma-Aldrich Inc., St. Louis, MO, USA) for ∼3 min and then rinsed with distilled water. For each assay, tissue was harvested from approximately 50 plants, as described previously [34,35]. Briefly, the seedlings were divided as follows: Stalk lower (base of Stalk 5 mm above the node); Stalk upper (top of Stalk 5 mm below the ligule of the first leaf); and leaf blades (L1 or L1 +  L2  +  L3). Stalk tissue comprised several concentric rings of leaves, the outermost being the first leaf sheath. L1 was the fully expanded blade, whereas L2 and L3 were still developing.

### 2.2. Isolation of Maize Organelles (Plastids and Mitochondria)

Organelles were isolated using high-salt buffer [HSB; 1.25 M NaCl, 40 mM HEPES pH 7.6, 2 mM EDTA pH 8, 0.1% BSA, 0.1% β-mercaptoethanol (Sigma-Aldrich Inc., St. Louis, MO, USA)] as described previously [34,35]. Tissues were homogenized in HSB, and differential centrifugation (Beckman Coulter, Brea, CA, USA) was applied to pellet plastids and mitochondria. The homogenate was centrifuged first at low speed (500× *g* for 5 min) to remove nuclei. The supernatant was then centrifuged at 3000× *g* for 10 min to pellet plastids. The remaining supernatant was centrifuged at 20,000× *g* for 15 min to pellet mitochondria. The plastid and mitochondria pellets were washed three times with chloroplast dilution buffer (CDB; 0.33 M D-sorbitol, 20 mM HEPES pH 7.6, 2 mM EDTA, 1 mM MgCl_2_, 0.1% BSA) and mitochondria dilution buffer (MDB; 0.4 M D-sorbitol, 0.1 M HEPES pH 7.6, 2 mM EDTA, 1 mM MgCl_2_, 0.1% BSA), respectively. The organelles were further purified using discontinuous (step) Percoll gradients (Sigma-Aldrich Inc., St. Louis, MO, USA) as described previously [34,35]. Finally, the purified mitochondria and plastids were resuspended and stored in a small volume of MDB or CDB, respectively.

### 2.3. Isolation of Organelle Proteins 

The organellar proteins were isolated as described [35,36]. Plastid and mitochondrial pellets were mixed with protein extraction buffer [50 mM HEPES, 1% CHAPS, 1x HALT protease inhibitor cocktail (Thermo Fisher Scientific; Waltham, MA USA)]. The mixture was vortexed for 5 min at high speed and was centrifuged at 10,000× *g* for 10 min (Eppendorf, Enfield, CT, USA). The supernatant was collected, and the total protein amount was quantified with a Pierce™ BCA Protein Assay kit (Thermo Fisher Scientific, Waltham, MA, USA).

### 2.4. Isolation of Organelle DNA 

Plastid DNA (ptDNA) and mitochondrial DNA (mtDNA) were extracted using cetyltrimethylammonium bromide (CTAB) with some modifications [35,37]. An equal volume of 2  ×  CTAB buffer [2% CTAB (*w*/*v*), 100 mM Tris/HCl (pH 8.0), 20 mM EDTA, 1.4 M NaCl, 1% polyvinylpyrrolidone (Mr 40,000; *w*/*v*) (Sigma-Aldrich Inc., St. Louis, MO, USA), preheated to 65 °C, and containing proteinase K (20 μg/mL)] (Sigma-Aldrich Inc., St. Louis, MO, USA) was added to the resuspended plastids or mitochondria and the mixtures incubated at 65 °C for 1 h. Then, 0.1 M phenylmethylsulfonyl fluoride (Sigma-Aldrich Inc., St. Louis, MO, USA) was added, followed by incubation at room temperature for 1 h. Then, RNase A (Thermo Fisher Scientific, Waltham, MA, USA) was added to 100 μg/mL to CTAB buffer with NaCl concentrations of 1.4 M so that it cleaves only ssRNA, and the samples were kept at 60 °C for 15 min. Next, potassium acetate (Sigma-Aldrich Inc., St. Louis, MO, USA) was added to 400 mM, and the mixtures were kept on ice for 15 min before centrifugation at 12,000× *g* for 10 min at 4 °C. Equal volumes of chloroform/isoamyl alcohol (24:1) were added to the recovered supernatant. The tubes were shaken, centrifuged at 12,000× *g* for 1 min, and the upper aqueous layer containing DNA was removed. The DNA was precipitated with two volumes of 100% ethanol (Sigma-Aldrich Inc., St. Louis, MO, USA) overnight at −20 °C before pelleting. DNA pellets were washed three times with 70% ethanol, dried, and then resuspended in TE buffer. Quantitation was performed using the Quant-IT DNA quantitation kit (Thermo Fisher Scientific, Waltham, MA, USA).

### 2.5. Measurement of RNase H Protein Levels Using Slot-Blot Assays

RNase H1 and RNase H2 protein slot-blot assays were performed using the Hoefer^®^ Slot blot filtration manifold (Thermo Fisher Scientific, Waltham, MA USA) as per the manufacturer’s instructions. The slot blot assays were performed as described in [30]. Briefly, plastid or mitochondrial proteins (50 µg) were blotted onto a nitrocellulose membrane (Millipore Sigma, Burlington, MA, USA), and preincubated in a 1x Tris-Buffered Saline (TBS) kit (Thermo Fisher Scientific, Waltham, MA, USA). Equal loading of the total proteins in samples was assessed by staining the membranes with Ponceau S, 0.2% *v*/*v* soln. in 5% acetic acid (Thermo Fisher Scientific, Waltham, MA USA) as described [35]. The membranes were blocked overnight at 4°C with 0.1% Tween 20 and 5% BSA (Sigma-Aldrich Inc., St. Louis, MO, USA) in 1x TBS. Membranes were incubated with either anti-RNase H1 (ab229078; rabbit polyclonal) or anti-RNase H2A (ab92876; mouse polyclonal) antibodies (1:1000 dilution; Abcam, Cambridge, UK). Both antibodies were raised against human RNases and share amino acid similarity with plant proteins [9,26]. After incubation for 2 h at 20 °C, membranes were washed three times with 1x TBST (TBS  +  0.01% tween 20) and followed by incubation for 1 h with horseradish peroxidase coupled anti-mouse (for RNase H2A) or anti-rabbit (for RNase H1) antibodies (1:10,000 dilution; Abcam, Cambridge, UK). For signal detection, the membrane was washed five times with 1x TBS-T and developed with 1-Step™ Ultra TMB-Blotting Solution (Thermo Fisher Scientific, Waltham, MA, USA). The signal intensity of each spot was quantified using ImageJ software (NIH, Bethesda, MA, USA). 

### 2.6. R-Loop Detection in Organelle DNA

R-loop detection in organelle DNA was performed as described in [38] with some modifications. PtDNA and mtDNA were spotted (50 ng per slot) onto N+ nylon membranes, with one for the S9.6 antibody and the other for the dsDNA antibody (Millipore Sigma, Burlington, MA, USA). The membranes were air dried for 2 min and then crosslinked using a UV crosslinker (Thomas Scientific, Swedesboro, NJ, USA) with an “Auto Crosslink” setting (1200 μJ × 100). After incubating the membrane in blocking solution (5% milk in Tris-buffered saline with 0.05% Tween-20 [TBST]) (Thermo Fisher Scientific, Waltham, MA USA) for 1 h at room temperature on a shaker (Benchmark scientific, Sayreville, NJ, USA), the blots were probed with either S9.6 R-loop (1:1000) (mouse monoclonal; Kerafast, Boston, MA, USA) or dsDNA (1:10,000) (mouse monoclonal; Abcam, Cambridge, UK) antibodies and later with secondary anti-mouse-AP (1:5000) (Abcam, Cambridge, UK) antibodies. The blots were developed with NBT/BCIP reagents (Roche, Indianapolis, IN, USA). The signal intensity of each spot was quantified using ImageJ software (NIH, Bethesda, MA, USA). The ratio of the integrated density of the R-loop/dsDNA signal was normalized by the tissue with the lowest value, which is set at one.

### 2.7. Ribonucleotide Detection in Organelle DNA

Ribonucleotides in organelle DNA were detected as described [39], using a modification of the nick-translation procedure. Typical nick-translation uses DNase to produce “single-stranded” nicks in dsDNA, then fill in the gaps with DNA pol I and radioactively labeled dNTPs. For these assays, the use of DNase was omitted, and instead, either RNase H1 or RNase H2 (Thermo Fisher Scientific, Waltham, MA, USA) was used to remove rNMPs that would be associated in RNA-DNA hybrid regions of the orgDNA. The ptDNA and mtDNA were isolated and mixed with either RNase H1, RNase H2, or only buffer (1x Thermopol; Thermo Fisher Scientific, Waltham, MA, USA), and the resulting gaps were then filled in using DNA Pol I (Sigma-Aldrich Inc., St. Louis, MO, USA), unlabeled dNTPs, and DIG-dUTP (Roche, Indianapolis, IN, USA). In addition, orgDNA without RNase treatment was also labeled using DNA Pol I and DIG-dUTP as a control and reference since it is known that endogenous orgDNA contains single-strand DNA gaps [40]. The DIG-labeled orgDNAs (2–4 µL) were spotted on the positively-charged nylon membranes (Millipore Sigma, Burlington, MA, USA) and probed with anti-DIG primary antibodies (1:1000) and secondary anti-rabbit-AP (1:1000) antibodies (Roche, Indianapolis, IN, USA). The signal on the blots was visualized using NBT/BCIP reagent (Roche, Indianapolis, IN, USA), and the integrated signal density of each dot blot was determined using ImageJ (NIH, Bethesda, MA, USA). The graphs show the integrated density relative to the tissue with the lowest value, which is set at one.

### 2.8. Statistical Analysis

Statistical analysis was performed as described in [34]. All assays were performed at least three times with similar results. The values in each figure are shown as mean relative values ± SE from three independent assays (biological replicates). Statistically significant differences between tissues were assessed by the ANOVA and Tukey honest significant difference test and are shown as asterisks, where * *p*-value ≤ 0.05, ** *p*-value ≤ 0.01, *** *p*-value ≤ 0.001, and *p*-values > 0.05 are indicated on the graphs. All the statistical analyses were performed using the Kaleidagraph 4.5 version for Windows (Synergy Software, Reading, PA, USA). The standard error of the ratio (i.e., the mean relative level) was determined using the following equation:$$\mathrm{SE}\left(\frac{x}{y}\right)=\left(\frac{x}{y}\right)\sqrt{{\left(\frac{\mathrm{SE}\left(x\right)}{x}\right)}^{2}}+{\left(\frac{\mathrm{SE}\left(y\right)}{y}\right)}^{2}$$
where x, SE(x), y, and SE(y) are the mean values. 

## 3. Results

There is a gradient in cell and organellar development from the basal meristem of the stalk to the tip of the maize leaf [41,42]. Previously, structural changes in orgDNA molecules during maize development were reported, including light-induced degradation and damage caused by reactive oxygen species (ROS) and glycating agents [34,35,43,44,45,46]. In this study, ribonucleotide damage was measured at three stages of maize development: Stalk lower (the base of the stalk), Stalk upper (top of the stalk), and the expanded leaf blade(s). L1 refers to the first and oldest leaf. L2 and L3 refer to the second and third leaves, respectively. In addition, leaf blade tissue from maize plants grown under light or dark conditions was evaluated for ribonucleotide damage in the orgDNA. The tissue with the lowest value in each set of assays was used as the baseline for comparison with other tissues and is set at 1.

### 3.1. Changes in the Level of R-Loops in orgDNA during Leaf Development and for Light-Grown Compared to Dark-Grown Plants

One typical method to assess RNA–DNA hybrid structures, such as R-loops, uses the S9.6 antibody. Possible artifacts due to double-stranded RNA cross-reactivity have been noted, however, particularly with immunofluorescence microscopy [47]. Ramiez et al. demonstrated that R-loop specificity can be achieved using dot-blots with purified DNAs treated with RNase T1 or RNase III [38]. They also showed that the S9.6 antibody does not bind with dsRNA or with dsDNA oligonucleotides [38]. Here, the S9.6 antibody was used to assess R-loops in dot-blots of purified maize organellar DNAs (plastid DNA, ptDNA; and mitochondrial DNA, mtDNA) that were pretreated with RNase A in high-salt conditions to specifically cleave ssRNA, but not dsRNA or the RNA strand in RNA–DNA hybrids (Section 2). Thus, if our orgDNA samples contained some dsRNAs, these would not be recognized by the S9.6 antibody [38]. Equal amounts of orgDNAs were spotted on the membrane, which was then probed with an S9.6 antibody. As a control, a second membrane was probed with a dsDNA antibody for normalization. Signal intensities (integrated densities from the S9.6 and dsDNA antibodies) were determined by ImageJ. Signal intensities of orgDNAs were normalized to those in mock samples (no orgDNA).

Changes in the amounts of R-loops were assessed during maize seedling development using S9.6 antibodies and dot-blot assays with orgDNAs isolated from leaf and stalk tissues (Figure 1). The data showed differences among the three tissues in the relative amounts of R-loops in both ptDNA and mtDNA. In plastids, DNA from the fully developed L1 leaf had approximately 2.7 times the R-loops found in Stalk lower (Figure 1a). The ratio was 2.2 for Stalk upper tissue compared to L1. In mitochondria, the corresponding ratios were 2.4 and 1.7, respectively (Figure 1b).

Previously, we found that orgDNA maintenance is influenced by responses to light signals: light not only leads to the greening of seedling leaves but also triggers the demise of both ptDNA and mtDNA in maize [34,44,46,48]. Here, R-loops were quantified in the DNA of plastids and mitochondria from light- and dark-grown total leaf tissues (L1 + L2 + L3). PtDNA from light-grown leaves had 2.4-fold more R-loops than dark-grown leaves (Figure 2a), while the level of R-loops was 1.5 times higher in mtDNA from light-grown leaves than dark-grown leaves (Figure 2b).

We conclude that the levels of R-loops in maize orgDNAs increase during leaf development, as reported previously for orgDNA damage caused by oxidative and glycation reactions. Persistent R-loops may therefore contribute to the demise of orgDNAs during the normal development of maize seedlings.

### 3.2. Changes in the Level of rNMPs in orgDNA during Leaf Development and for Light-Grown Compared to Dark-Grown Plants

Ribonucleotide monophosphates (rNMPs) are typically incorporated into DNA during replication and are then replaced by deoxyribonucleotides. We quantified the rNMPs embedded in orgDNA using the method described in [39] with some modifications (see Materials and Methods). Briefly, RNase H1 or RNase H2 was used to remove ribonucleotides, creating discontinuities in the orgDNA strands. Then, the resulting gaps were filled in using DNA Pol I and DIG-dUTP. As a control, orgDNAs were treated with buffer without RNase H1 or H2. Next, the amounts of DIG-dUTP in orgDNAs were measured as the signal intensity on dot blots following incubation with anti-DIG-antibodies and detection with the NBT/BCIP reagent. Signal intensities were normalized to buffer-only samples. We found 4.4 times more rNMPs (after RNase H1 treatment) and 5.6 times more rNMPs (after RNase H2 treatment) in ptDNA from L1 than from Stalk lower (Figure 3a,c). In mitochondria, the corresponding ratios were 4.2 and 4.4, respectively (Figure 3b,d). For both ptDNA and mtDNA, the rNMP increases were smaller for the Stalk upper tissues, as might be expected since development proceeds from Stalk lower to Stalk upper to L1. 

The orgDNAs from light-grown and dark-grown leaves were isolated and treated with RNase H1 or RNase H2 to remove rNMPs. The repaired nicks were then quantified to assess the level of rNMPs. For ptDNA, the data show that in comparison to dark-grown leaves, light-grown leaves had 3.4 times (+RNase H1) and 6.8 times (+RNase H2) more rNMPs (Figure 4a,c). For mtDNA, rNMP levels in light-grown leaves were 1.3 and 2.8 times higher than in dark-grown leaves after RNase H1 and RNase H2 treatments, respectively (Figure 4b,d).

We conclude that rNMPs accumulate in both orgDNAs during maize leaf development. Persistent rNMPs may therefore contribute to the decline of orgDNAs during the normal development of maize seedlings.

### 3.3. Changes in the Level of Organellar RNase H during Leaf Development and for Light-Grown Compared to Dark-Grown Plants 

The RNase H enzymes (RNase H1 and H2) eliminate the RNA moiety from RNA–DNA hybrids and function in reducing ribonucleotides and R-loops. Since the levels of rNMPs and R-loops in orgDNA from L1 were higher than those from Stalk lower and Stalk upper tissues, this difference may be attributed to lower levels of RNase H in L1. Thus, we measured the levels of both RNase H1 and RNase H2A protein in organelles (Materials and Methods). This was performed using slot blot assays and anti-RNase H1 or anti-RNase H2 antibodies. For RNase H1 and H2A, the antibodies were raised against proteins from humans, which show approximately 50–60% similarity with Arabidopsis RNase H1 and H2A [31,49]. Plastids isolated from Stalk lower had 1.3 times more RNase H1 and 1.5 times more RNase H2A than L1 (Figure 5a, c). Similarly, mitochondria isolated from Stalk lower had 1.4 times more RNase H1 and 1.7 times more RNase H2A than L1 (Figure 5b,d).

Organellar proteins were isolated from light- and dark-grown leaves and used in slot blot assays with RNase H antibodies. Equal amounts of total protein were compared (Ponceau S staining; Materials and Methods). The RNase H1 level was 1.2 times higher in isolated plastids from dark-grown leaves than light-grown leaves (Figure 6a), while the RNase H2A level was 1.4 times higher in plastids from dark-grown leaves than light-grown leaves (Figure 6c). In isolated mitochondria, there was 1.4 times more RNase H1 and 1.8 times more RNase H2A from dark-grown leaves compared to light-grown leaves (Figure 6b,d).

We conclude that the levels of RNase H proteins decrease during maize leaf development and that a lack of repair/removal of rNMPs results in increased ribonucleotide damage in orgDNAs within the mature leaf tissue. Although RNase H1 has been reported in Arabidopsis plastids and mitochondria from various organisms, to our knowledge, none of the RNase H2 protein subunits (A, B, or C) or RNase H2 activity has been previously documented in plant plastids or mitochondria.

## 4. Discussion

We previously reported that the levels of two types of molecular-damaging agents, ROS and glycation, increase when maize plants develop from the basal meristem to the green leaf [2,34,35]. R-loops and rNMPs in DNA are formed normally during DNA replication and transcription, after which they are promptly removed by RNase H. However, if they persist, these aberrant forms may lead to genome instability and represent a third type of damage to DNA. Here, we report on that third type of damage to orgDNA. As damage increases during leaf development, damage defense decreases, as we found previously for both ROS and glycation. 

The levels of R-loops in orgDNA were measured during normal plant development using the S9.6 antibody, which recognizes RNA–DNA hybrid structures. Our data show a higher level of R-loops in the fully developed leaf (L1) compared to the base of the stalk. We also compared the R-loop levels between light- and dark-grown leaves. This comparison reveals useful information about orgDNA damage and repair processes. Dark-grown leaves have high orgDNA copy numbers and structural integrity. A rapid decline in orgDNA was observed when etioplasts developed into photosynthetically-active chloroplasts under light conditions [50]. A lower level of R-loops was found in orgDNA from dark-grown leaves than from light-grown leaves. A recent study compared dynamic patterns of R-loops in nuclear DNA for 53 Arabidopsis tissues from diverse developmental stages grown under different light and temperature conditions and in various environmental stimuli [50]. Those data showed that the changing distribution patterns of R-loops were necessary for the normal plant life cycle and heat-stress responses. Although the formation of R-loops can contribute to normal physiology, these structures can also cause DNA damage and genome instability, as shown in other organisms [50,51]. 

As DNA transcription and replication use the same template, R-loops represent a barrier to DNA replication. ROS can generate R-loops leading to single- and double-strand breaks in transcribed regions. The role of ROS in transcription-coupled homologous recombination (TC-HR) in human and yeast cells has been investigated and has shown that ROS activates TC-HR at a transcriptionally active locus, implying that TC-HR is involved in repairing ROS-induced DNA damage in transcribed regions [52]. In addition, oxidative stress destabilizes the mitochondrial genome via unscheduled R-loop accumulation by impairing the recruitment of RNAse H1 to the regulatory regions of human mtDNA [53,54]. Harmful R-loops are most likely to accrue when their replacement rate is impeded, perhaps due to low levels of RNase repair systems. R-loop-induced DNA damage and genome instability are thought to be caused primarily by transcription–replication conflicts [55].

Various means ensure that DNA is faithfully passed to daughter cells at cell division. These are employed both during and after replication and include DNA repair and the use of high-fidelity DNA polymerases with proofreading activities in nuclei and organelles [5,56,57]. Although the proofreading function of DNA polymerases can eliminate rNMPs [8,58], this is insufficient to stop considerable levels of rNMPs from being integrated into the DNA backbone. Multiple rNMPs are more detrimental to cell viability than a single rNMP, and replicative DNA polymerases stall at four (or even fewer) consecutive rNMPs [8,59].

We measured the levels of rNMPs in orgDNA by removing rNMPs with RNase H1 and RNase H2 enzymes and using DIG-dUTP to fill in the gaps. Our data show higher levels of ribonucleotides (a run of at least four with RNase H1 or one with RNase H2) in organelles from L1 than in stalk lower tissue. When comparing rNMPs between orgDNAs from light- and dark-grown leaves, we found that light-grown leaves had relatively higher levels of rNMPs than dark-grown leaves. Interestingly, higher levels of rNMP removal by RNase H2 than H1 suggest that single rNMPs are more frequently incorporated in the orgDNAs than a sequence of four or more rNMPs.

Most misincorporated rNMPs are removed by the evolutionarily-conserved type 2 ribonuclease H-dependent ribonucleotide excision pathway to maintain genome integrity. DNA sequences for three RNase H1-like proteins, AtRNH1A, -B, and -C, have been found in the Arabidopsis genome that show similar domain structures to mammalian RNase H1 [28]. While the in vivo activities of AtRNH1A in the nucleus are yet unknown, AtRNH1B and AtRNH1C regulate R-loop homeostasis in mitochondria and chloroplasts, respectively [28,31,59]. RNH1A, RNH1B, and RNH1C share extensive amino acid sequence similarity and activity characteristics with human RNase H1 [31]. The levels of RNase H1 proteins in maize organelles were assessed using a human RNAse H1 antibody. Based on slot-blot analyses using this RNase H1 antibody, we found a higher level of RNase H1 in the organelles from Stalk lower and dark-grown leaves than L1 and light-grown leaves, respectively. 

In *E. coli*, the RNase HII protein functions as a single polypeptide unit that “rides” along with RNA polymerase and is thus delivered to where ribonucleotides need to be removed [60]. In Arabidopsis, the RNase H2 holoenzyme comprises three polypeptide units and was found to be important for suppressing nuclear genome instability, and an accumulation of rNMPs in nuclear DNA was noted for mutants that lack active RNase H2 [11]. In addition, sequence similarity for the RNase H2 subunit A was found among mice, humans, and some plants, including Arabidopsis and maize [11]. We used antibodies for the RNase H2A subunit from humans to assess the presence of RNase H2 in maize plastids and mitochondria. The slot blot data indicate the presence of RNase H2 subunit A in both organelles.

The level of RNase H2A showed the same trend as RNase H1: higher in Stalk lower and dark-grown leaves than L1 and light-grown leaves. We cannot confirm the presence of RNase H2 activity in maize organelles based only on the presence of one of (presumably) three subunits of RNase H2. Neither RNase H2 protein nor enzymatic machinery has been found in mitochondria from humans and yeast [5,61], although RNase H2 has been found in mitochondria from the trypanosomatid protists *Leishmania* and *Trypanosoma* [62]. Further research is required to show whether any RNase H2 enzymatic activity is present in maize organelles. 

Based on our data, we hypothesize that higher levels of RNase H protect orgDNAs in Stalk lower and dark-grown leaves from ribonucleotide/R-loop damages, and the lower levels of these enzymes in light-grown leaf tissues may result in greater orgDNA damage due to unrepaired ribonucleotides/R-loops. R-loops and rNMPs in DNA are produced during normal transcription and DNA replication. They can also arise from genotoxic perturbations (mutations, toxins, and extremes of temperature, salinity, and radiation). Unless removed, persistent R-loops and rNMPs are one form of “damage” to DNA. Our data reveal that such damage to orgDNAs increases during the normal development (without imposed perturbations) of maize leaves from slow-growing colorless cells in the basal meristem to rapidly expanding cells in the green leaf. During this developmental transition, the levels of the primary agents of defense (RNase H1 and H2) decrease. This decrease is but one facet of a general trend in maize. As damage from R-loops/rNMPs, ROS, and glycation increases during development, the “effort” to repair the damage decreases, and the orgDNAs are abandoned.

## 5. Conclusions and Outlook

As ribonucleotide damage to orgDNA increases during maize development from meristem to leaf blade, the levels of the repair agent, RNase H, decrease. This seemingly paradoxical abandonment of orgDNA was noted previously for both oxidative and glycation damage [2]. How might this damage response process be regulated? One idea stems from the 3-subunit nature of RNase H2 and a recent proposal for how the single-subunit RNase HII in bacteria is targeted to sites in need of repair [63]. Perhaps each subunit type directs the few copies per cell of nucleus-encoded RNase H2 to its cognate organelle. Differential expression of the 3 subunits could allow repair of orgDNA damage in the meristem, but not green leaves, of maize and is a testable hypothesis.

## Figures and Tables

**Figure 1 plants-12-03161-f001:**
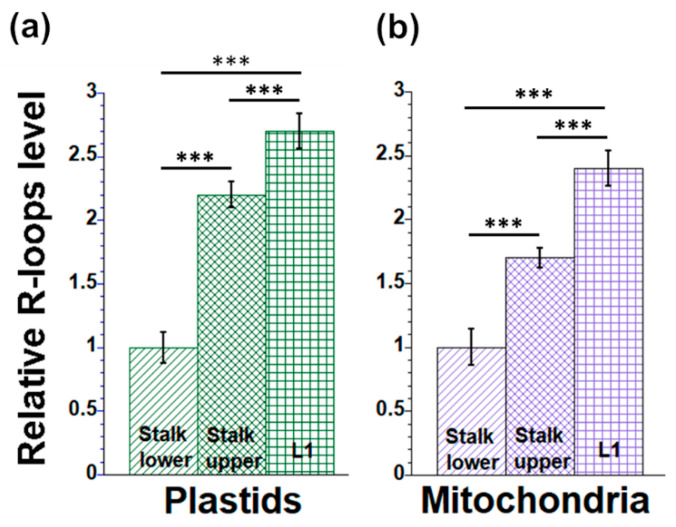
R-loop levels in orgDNA during maize development. Plastid (**a**) and mitochondrial (**b**) DNAs from Stalk lower (S1), Stalk upper (S2), and L1 (first leaf) were isolated, spotted onto nylon membranes, and probed with R-loop antibodies. Signal intensity was measured following detection using NBT/BCIP reagents. The orgDNA was assayed using dsDNA antibodies and similar detection procedures. The integrated density of each dot blot was determined using ImageJ. The ordinate shows the R-loop/dsDNA signal normalized to the tissue with the lowest value (Stalk lower), which is set at one. For the data in Figure 1, Figure 2, Figure 3, Figure 4, Figure 5 and Figure 6, all assays were performed at least three times. The statistically significant differences were measured using ANOVA statistic test with post hoc analysis using Tukey’s HSD and are shown as asterisks, where *** *p*-value ≤ 0.001.

**Figure 2 plants-12-03161-f002:**
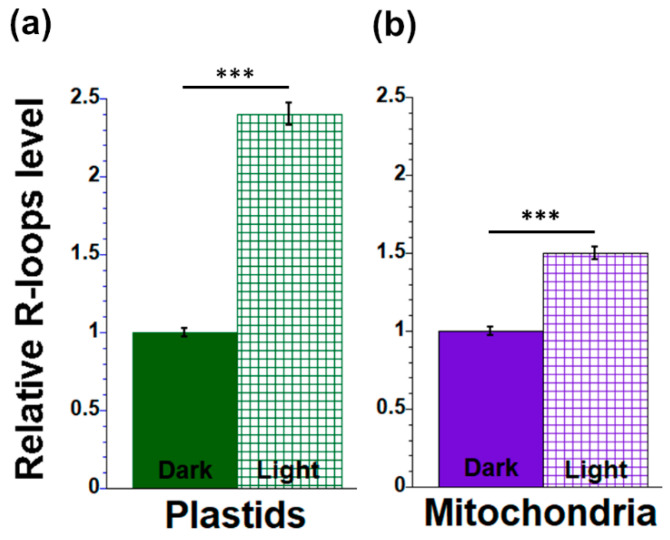
R-loop levels in orgDNA from maize seedling grown under light and dark conditions. Plastid (**a**) and mitochondrial (**b**) DNAs from total leaf tissues of seedlings grown in light and dark were isolated and spotted onto membranes. The membranes were probed with R-loop antibodies and signal intensity was measured following detection using NBT/BCIP reagents. OrgDNA in each dot was determined using dsDNA antibodies and similar detection procedures. The integrated density of each dot blot was determined using ImageJ. The ratio of integrated density of R-loop/dsDNA signal was normalized to the tissue with the lowest value (Dark-grown leaves), which is set at one. The statistically significant differences were measured using ANOVA statistic test with post hoc analysis using Tukey’s HSD and are shown as asterisks, where *** *p*-value ≤ 0.001.

**Figure 3 plants-12-03161-f003:**
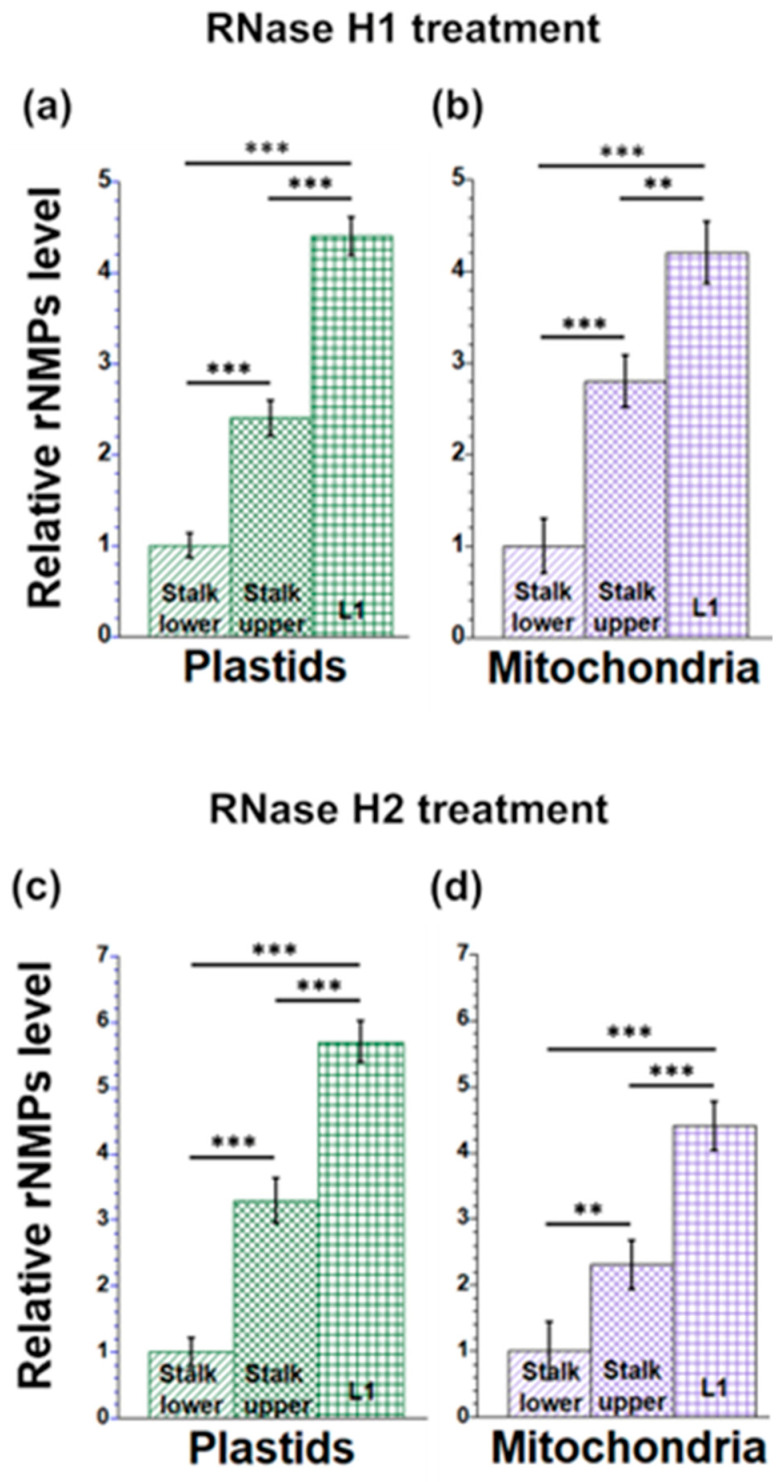
rNMP levels in orgDNA during maize development. Plastid (**a**,**c**) and mitochondrial (**b**,**d**) DNAs from Stalk lower, Stalk upper, and L1 were isolated and treated with RNase H1 (**a**,**b**) and RNase H2 (**c**,**d**) to remove ribonucleotides at RNA/DNA hybrid regions. Labeling of regions where rNMPs were removed was carried out using DNA Pol I and DIG-dUTP, and this was also conducted for orgDNA without RNase H treatment. The ratio of the integrated density of +RNase H/–RNase H signal was normalized to the tissue with the lowest value (Stalk lower), which is set at one. The statistically significant differences were measured using ANOVA statistic test with post hoc analysis using Tukey’s HSD and are shown as asterisks, where ** *p*-value ≤ 0.01, *** *p*-value ≤ 0.001.

**Figure 4 plants-12-03161-f004:**
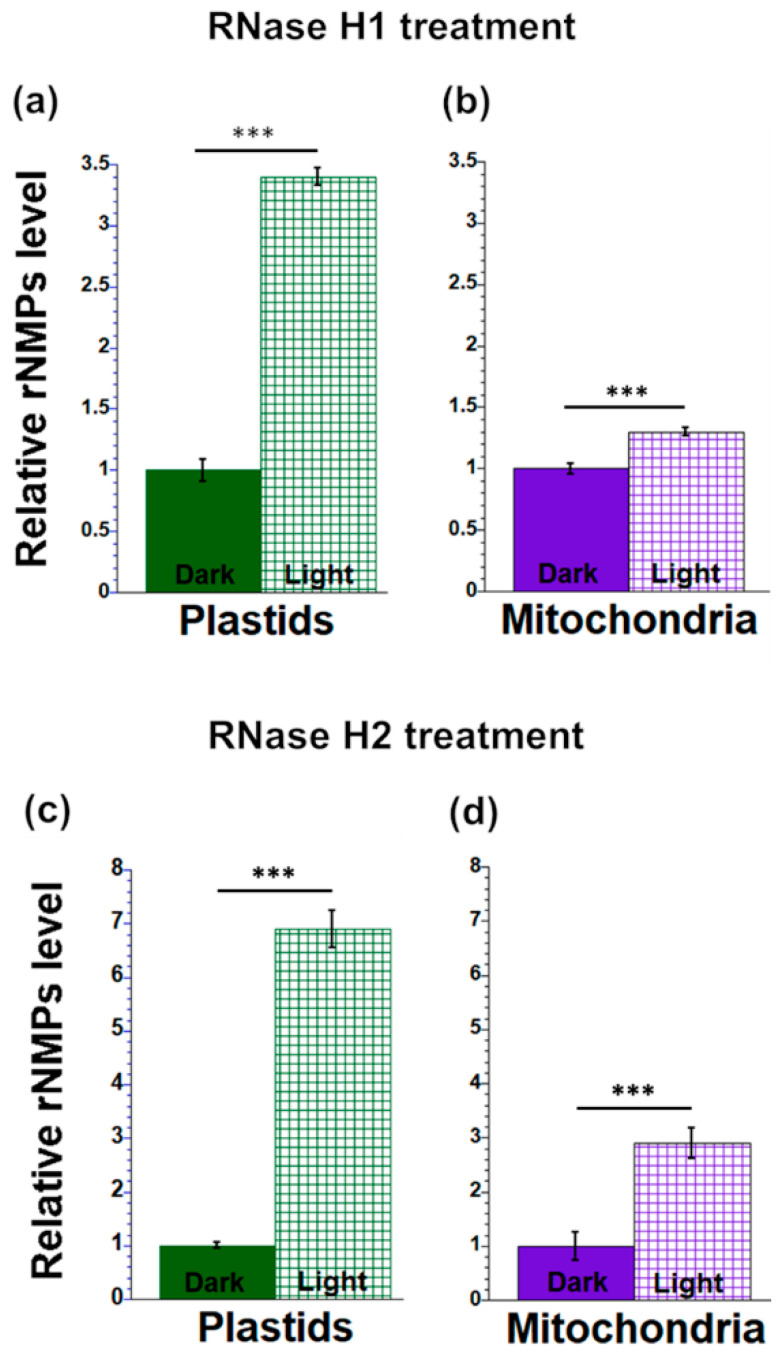
rNMP levels in orgDNA from maize seedling grown under light and dark conditions. Plastid (**a**) and mitochondrial (**b**) DNAs from total leaf tissues of seedlings grown in light and dark were isolated and treated with RNase H1 (**a**,**b**) or RNase H2 (**c**,**d**) to remove ribonucleotides at RNA/DNA hybrid regions (as described for Figure 3). The integrated density of each dot blot was determined using ImageJ. The ratio of +RNase H/–RNase H signal was normalized to the tissue with the lowest value (dark-grown leaves). The statistically significant differences were measured using ANOVA statistic test with post hoc analysis using Tukey’s HSD and are shown as asterisks, where *** *p*-value ≤ 0.001.

**Figure 5 plants-12-03161-f005:**
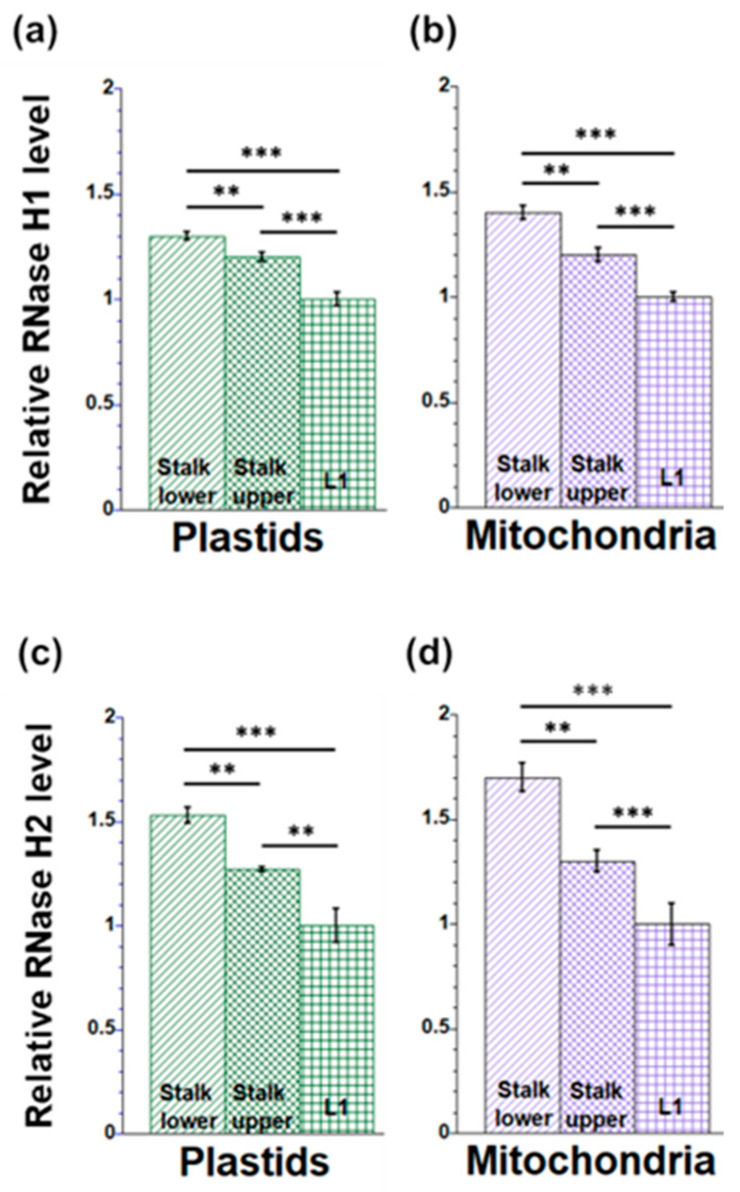
Measurement of RNase H1 (**a**,**b**) and RNase H2A (**c**,**d**) protein levels in organelles during maize development. Plastid (**a**) and mitochondrial (**b**) proteins from Stalk lower, Stalk upper, and L1 were isolated and spotted to nitrocellulose membranes, and then probed with either RNase H1 (**a**,**b**) or RNase H2A (**c**,**d**) antibodies. The ratio of integrated signal density was normalized to the tissue with the lowest value (L1), which is set at one. The statistically significant differences were measured using ANOVA statistic test with post hoc analysis using Tukey’s HSD and are shown as asterisks, where ** *p*-value ≤ 0.01, *** *p*-value ≤ 0.001.

**Figure 6 plants-12-03161-f006:**
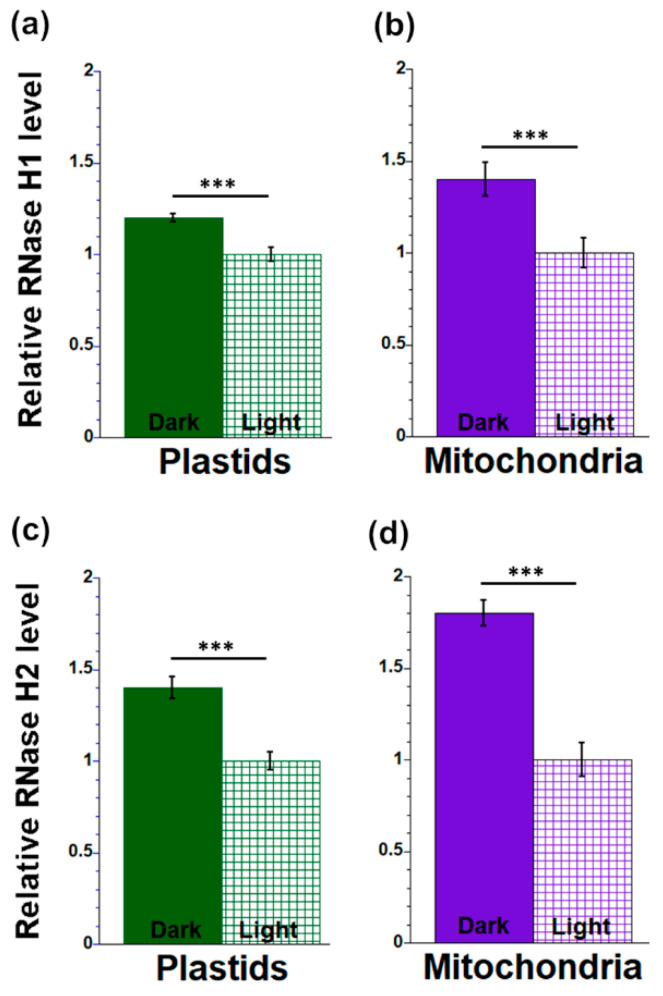
Measurement of RNase H1 (**a**,**b**) and RNase H2A (**c**,**d**) protein levels in organelles from maize seedlings grown under light and dark conditions. Plastid (**a**,**c**) and mitochondrial (**b**,**d**) proteins from total leaf tissues of seedlings grown in light and dark were isolated and probed with RNase H1 (**a**,**b**) and RNase H2A (**c**,**d**) antibodies. The signals were normalized to the tissue with the lowest value (dark-grown leaves), as in Figure 5. The statistically significant differences were measured using ANOVA statistic test with post hoc analysis using Tukey’s HSD and are shown as asterisks, where *** *p*-value ≤ 0.001.

## Data Availability

Not applicable.

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
