# Peer review of "Ribonucleotide and R-Loop Damage in Plastid DNA and Mitochondrial DNA during Maize Development"

_plants, 2023, doi:10.3390/plants12173161_

Round 1

Reviewer 1 Report

Ribonucleotides are abundant contaminating nucleotides in DNA and have important biological implications. The authors investigate changes in the levels of ribonucleotides and R-loops in the plastid and mitochondrial DNA of maize and of RNase H proteins in organelles during leaf development and in leaves grown in light and dark conditions. The study relies mainly on the signal from the S9.6 antibody to conclude the presence of R-loops and to compare their levels in different leaf tissues of maize. However, the potential off-target effects of S9.6 antibodies were not acknowledged or discussed, which are documented in the literature (J. Cell Biol. 2021 Vol. 220 No. 6 e202004079). In addition, experiments, such as qPCR assays, documenting the purity of organellar DNA are lacking. Without these important validations, the results are preliminary. 

Other minor points: 

In Introduction, it would be beneficial to discuss the known and unknowns regarding the presence of ribonucleotides and other modifications in the plastid and mitochondrial DNA of maize, especially in the comparison with the known info of the Arabidopsis genome. 

In discussion, it would be more meaningful to discuss the role of RNase H enzymes in the context of what is known with the model organism Arabidopsis. 

none

Author Response

The authors' comments are in bold -

  1. Ribonucleotides are abundant contaminating nucleotides in DNA and have important biological implications. The authors investigate changes in the levels of ribonucleotides and R-loops in the plastid and mitochondrial DNA of maize and of RNase H proteins in organelles during leaf development and in leaves grown in light and dark conditions. The study relies mainly on the signal from the S9.6 antibody to conclude the presence of R-loops and to compare their levels in different leaf tissues of maize. However, the potential off-target effects of S9.6 antibodies were not acknowledged or discussed, which are documented in the literature (J. Cell Biol. 2021 Vol. 220 No. 6 e202004079).

The S9.6 antibody that was used to detect R-loops may have off-target effects and detect ssRNA or dsRNAs instead of only RNA-DNA hybrids. In the results section (lines 268-278, page 8) we added information and references concerning possible cross-reactivity, such as when performing immunofluorescence microscopy. In this study, we used the dot blot method by Ramirez et al., 2022. They clearly show that S9.6 antibody does not cross react with dsRNA in dot blot assay using purified DNA.

  1. In addition, experiments, such as qPCR assays, documenting the purity of organellar DNA are lacking. Without these important validations, the results are preliminary.

We do not quite understand what the reviewer means by using qPCR as test for purity of maize orgDNAs. For many years, the research focus in our lab has been on plant plastid and mitochondrial DNAs and during that time we have used various methods to verify the isolation of “pure” organelles and organellar DNAs, including gel electrophoresis, blot hybridization, fluorescence microscopy, and PCR. Also, the use of PCR with maize (and other plant) genomes can be complicated due to the well-documented transfer of organellar genome sequences into the nuclear genome, as we described in Kumar et al. (2011) Curr Genet DOI 10.1007/s00294-011-0342-6. Thus, our data are not “preliminary” and accurately represent the properties of orgDNA isolated from the plants.

  1. In Introduction, it would be beneficial to discuss the known and unknowns regarding the presence of ribonucleotides and other modifications in the plastid and mitochondrial DNA of maize, especially in the comparison with the known info of the Arabidopsis genome.
  2. In discussion, it would be more meaningful to discuss the role of RNase H enzymes in the context of what is known with the model organism Arabidopsis.

We agree to this idea and have added additional words and modified text to this effect as noted below:

lines 112-121, page 4 (Introduction)

lines  451-469, page 15 (Discussion)

lines 490-499, page 16 (Conclusions)

Reviewer 2 Report

The manuscript is well presented and address an important area of research. The authors need to present 'Conclusions' at the end of the manuscript highlighting the salient findings of the study.

Author Response

The authors' comments are in bold -

  1. The authors need to present 'Conclusions' at the end of the manuscript highlighting the salient findings of the study.

We added a Conclusions section (lines 490-499, page 16)

Reviewer 3 Report

Reviewer Reports:

I recommend a major amendment at this level.

General comments:

The manuscript entitled “Ribonucleotide and R-loop damage in plastid DNA and 1 mitochondrial DNA during maize development” was reviewed. The work carried out in the manuscript is interesting and aimed at the third type of orgDNA damage (ribonucleotides 94 and R-loops) and damage defense (RNase H) during leaf development. Better connect your research findings to previous works published in Plants and in other top journals. The innovation and the importance of this work are not clearly highlighted in the abstract, introduction, and conclusions. Please work on this and prove to us why this work is valuable. Additionally, the novelty of the research still is not clear and the discussion and conclusions can not satisfy me. Please also remove ANY lumped references. Please define each of them separately to avoid inappropriate citations. It is better to do not to use the first-person's pronoun. Do not use "we, us, or our" throughout the paper. It is recommended that the authors work with a science editor who is proficient in the Native English language to improve the organization and delivery of some portions of the manuscript. Too many abbreviations are used in the analysis and results. I recommend a nomenclature section for the abbreviations and variables used throughout the passage. Please provide research highlights. Please provide also a graphical abstract to provide a visual summary of the main findings of the study. Please embed the figures and the tables in the text, at least in the marked version.  The journal's author guidelines and instructions should be followed in preparing the revised version. Other main remarks that in my opinion needs attention are the following:

Detailed comments:

The abstract should state briefly the purpose of the research, the principal results, and major conclusions. In the abstract, please add an indication of the achievements from your study that are relevant to the journal's scope. Please be concise - maximum 1-2 lines. Quantitative information should be provided in the abstract. The abstract should include a sentence about your findings, discussions, and conclusions in your abstract and underscore the scientific value added to your paper in your abstract.

The review of the literature needs more updating with works to have a clear and concise state-of-the-art analysis. This should more clearly show the knowledge gaps identified and link them to the paper's goals. The introduction section is poorly organized. While the general introduction is acceptable, the state-of-the-art review that follows is very difficult to understand and no specific thoughts can be inferred. The major defect of this study is the debate or argument is not clearly stated. You may see these articles and follow them in the revised version. The relevant reference may be of interest to the author according to below: https://doi.org/10.1016/j.envres.2023.116363; https://doi.org/10.1038/s41598-022-23816-3; Please eliminate the use of redundant words. Eg. In this way, Recently, Respectively, therefore, currently, thus, hence, finally, to do this, first, in order, however, moreover, nowadays, today, consequently, in addition, additionally, furthermore. Please revise all similar cases, as removing these term(s) would not significantly affect the meaning of the sentence. This will keep the manuscript as CONCISE as possible. Please check ALL. Avoid beginning or ending a sentence with one or a few words, they are usually redundant. Kindly revise all. What are the key findings of the research? How does the extent of ribonucleotide and R-loop damage change during maize development, and are there any specific patterns observed?

Please avoid having one heading after another with no discussion in between as in the case of Sections 2 and 2.1. Kindly inspect the entire document for similar instances and revise accordingly. Please add in the beginning your scientific hypothesis. In the course of describing the performed actions, please provide reader guidance, sufficient for understanding why those actions have been performed. The percentage purity and company of all reagents/chemicals utilized must be reported. Though some of the model/brands of the equipment used was stated, their country of manufacture should be reported as well.

The structure of this work should be reorganized. For example, the Section of results should be combined with the Discussion. The authors are suggested to have the results and discussion part together. All the findings of the current work need to be compared and discussed with the results of other researchers finding instead of having a general comparison with other researchers' works. The authors should perform a comparison between the forecasting results. In your discussion section, please link your empirical results with a broader and deeper literature review.  What are the potential implications of ribonucleotide and R-loop damage in plastid and mitochondrial DNA for maize development, and what effects might it have on plant growth and physiology? Did the study identify any factors that influence the accumulation of ribonucleotide and R-loop damage in plastid and mitochondrial DNA? Are there any environmental or genetic factors that may exacerbate or mitigate this damage?

Please make sure your conclusions section underscores the scientific value-added of your paper, and/or the applicability of your findings/results. Highlights the novelty of your study. In the conclusions, in addition to summarising the actions taken and results, please strengthen the explanation of their significance. It is recommended to use quantitative reasoning compared with appropriate benchmarks, especially those stemming from previous work. How does the study address potential limitations or biases in the experimental design, and what steps were taken to ensure the reliability and validity of the results? Based on the findings, what are the potential avenues for further research on ribonucleotide and R-loop damage in plant organelle DNA and its role in maize development?

Please check the reference section carefully and correct the inconsistency. Please update this section.

Need to improve during revision. 

Author Response

The author's comments are in bold -

The manuscript entitled “Ribonucleotide and R-loop damage in plastid DNA and 1 mitochondrial DNA during maize development” was reviewed. The work carried out in the manuscript is interesting and aimed at the third type of orgDNA damage (ribonucleotides 94 and R-loops) and damage defense (RNase H) during leaf development. Better connect your research findings to previous works published in Plants and in other top journals. The innovation and the importance of this work are not clearly highlighted in the abstract, introduction, and conclusions.

  • Please work on this and prove to us why this work is valuable. Additionally, the novelty of the research still is not clear and the discussion and conclusions can not satisfy me.

We modified the Introduction and Discussion sections and added conclusions

  • Please also remove ANY lumped references. Please define each of them separately to avoid inappropriate citations.

We removed the lumped references where needed, although grouping [is this what the reviewer means by “lumped references”?] is very commonly used in most published papers including several published articles in this journal, Plants, when several refs are needed to support a statement.

  • It is better to do not to use the first-person's pronoun. Do not use "we, us, or our" throughout the paper. It is recommended that the authors work with a science editor who is proficient in the Native English language to improve the organization and delivery of some portions of the manuscript.

Although we agree that “excessive” use of first-person pronouns should be avoided, occasional use is sometimes warranted to stress that this was work done in our lab. Nevertheless, we have tried to remove “we”, etc. throughout the manuscript where it is not needed.

  • Too many abbreviations are used in the analysis and results. I recommend a nomenclature section for the abbreviations and variables used throughout the passage.

We used the journal’s guideline for abbreviations. Here is what it says “Acronyms/Abbreviations/Initialisms should be defined the first time they appear in each of three sections: the abstract; the main text; the first figure or table. When defined for the first time, the acronym/abbreviation/initialism should be added in parentheses after the written-out form.”  We have followed these guidelines,  but if the editor wants us to add a glossary of abbreviations, we will do so.

  • Please provide research highlights. Please provide also a graphical abstract to provide a visual summary of the main findings of the study.

We are working on the graphical abstract. We plan to provide a graphical abstract once the manuscript is accepted for publication.

  • Please embed the figures and the tables in the text, at least in the marked version. The journal's author guidelines and instructions should be followed in preparing the revised version.

We inserted the figures into the main text close to their first citation (see lines 266-402, pages 8-14).

Detailed comments:

  1. The abstract should state briefly the purpose of the research, the principal results, and major conclusions. In the abstract, please add an indication of the achievements from your study that are relevant to the journal's scope. Please be concise - maximum 1-2 lines. Quantitative information should be provided in the abstract. The abstract should include a sentence about your findings, discussions, and conclusions in your abstract and underscore the scientific value added to your paper in your abstract.

We modified the abstract (lines 36-45, page 2). An abstract typically describes the principal findings in general terms but does not include quantitative data such as the fold increases or decreases. We have revised the Abstract but have not listed all the quantitative information that is clearly presented in the figures.

  1. The review of the literature needs more updating with works to have a clear and concise state-of-the-art analysis. This should more clearly show the knowledge gaps identified and link them to the paper's goals. The introduction section is poorly organized. While the general introduction is acceptable, the state-of-the-art review that follows is very difficult to understand and no specific thoughts can be inferred. The major defect of this study is the debate or argument is not clearly stated. You may see these articles and follow them in the revised version. The relevant reference may be of interest to the author according to below: https://doi.org/10.1016/j.envres.2023.116363; https://doi.org/10.1038/s41598-022-23816-3;

We modified the introduction section (lines 122-134, page 4). However, the Reviewers statements here are somewhat vague, and we are uncertain as to just what is suggested. The reference articles mentioned by reviewer are not relevant to our study.

  1. Please eliminate the use of redundant words. Eg. In this way, Recently, Respectively, therefore, currently, thus, hence, finally, to do this, first, in order, however, moreover, nowadays, today, consequently, in addition, additionally, furthermore. Please revise all similar cases, as removing these term(s) would not significantly affect the meaning of the sentence. This will keep the manuscript as CONCISE as possible. Please check ALL. Avoid beginning or ending a sentence with one or a few words, they are usually redundant. Kindly revise all. What are the key findings of the research? How does the extent of ribonucleotide and R-loop damage change during maize development, and are there any specific patterns observed?

We modified the text to indicate the key findings at the end of each Results section (see lines 266-402, pages 8-14).  We think that our text now provides what the Reviewer asks for in the last two sentences.  Again, however, the Reviewer’s statements are somewhat vague, so it is unclear exactly what the Reviewer wants. We respectfully disagree with the Reviewer’s comments about using adverbs such as however, therefore, etc. as often these do help clarification and provide continuity between statements.

  1. Please avoid having one heading after another with no discussion in between as in the case of Sections 2 and 2.1. Kindly inspect the entire document for similar instances and revise accordingly. Please add in the beginning your scientific hypothesis. In the course of describing the performed actions, please provide reader guidance, sufficient for understanding why those actions have been performed.

We have reduced the number of subsections in the Results to three, each with a brief summary, so as to avoid the “one heading after another” effect (see lines 266-402, pages 8-14).  We also modified the Introduction section (lines 122-134, page 4) to clarify why we began this research project that led to our discovery that the third type of damage (ribonucleotide damage) was “treated” by developing maize plants in the same way as the other two types of damage (oxidative and glycation).

  1. The percentage purity and company of all reagents/chemicals utilized must be reported. Though some of the model/brands of the equipment used was stated, their country of manufacture should be reported as well.

We added the relevant information (model/brands & country) in the M and M section (lines 138-253, pages 4-7)

  1. The structure of this work should be reorganized. For example, the Section of results should be combined with the Discussion. The authors are suggested to have the results and discussion part together. All the findings of the current work need to be compared and discussed with the results of other researchers finding instead of having a general comparison with other researchers' works. The authors should perform a comparison between the forecasting results.

We followed the journal’s guidelines and thus present the Results and Discussion separately. As for comparison with previously-published data, we found no such data concerning the developmental changes in organellar DNA for any plant. There are data for Arabidopsis (and these are described in lines 112-121, page 4 & lines 451-469, page 15). some of these data include R-loops mediated developmental changes and activity of RNAse H enzymes in Arabidopsis. There is not much. The Reviewer’s last sentence, “The authors…forecasting results“ is cryptic, and we cannot respond to this sentence.

  1. In your discussion section, please link your empirical results with a broader and deeper literature review. What are the potential implications of ribonucleotide and R-loop damage in plastid and mitochondrial DNA for maize development, and what effects might it have on plant growth and physiology? Did the study identify any factors that influence the accumulation of ribonucleotide and R-loop damage in plastid and mitochondrial DNA? Are there any environmental or genetic factors that may exacerbate or mitigate this damage?

In this study, we did not identify any factors (other than stage of development) that influence the accumulation of R-loops and rNMPs. Furthermore, we did not investigate the effect of R-loops/rNMPs on plant growth and physiology. Previous studies have suggested that R-loops are involved in multiple genome regulatory processes and may play important roles in plant growth and responses to environmental factors. However, it is still unknown how they regulate growth (https://doi.org/10.1093/jxb/erac433). In this study, our focus was to compare R-loop/rNMP levels among different tissues. We compared their levels during the normal course of maize development. As shown in one of our experiments, we observed that normal light conditions for growth may lead to accumulation of R-loops/rNMPs. Beyond this, we cannot comment on what the Reviewer is asking.

  1. Please make sure your conclusions section underscores the scientific value-added of your paper, and/or the applicability of your findings/results. Highlights the novelty of your study. In the conclusions, in addition to summarizing the actions taken and results, please strengthen the explanation of their significance. It is recommended to use quantitative reasoning compared with appropriate benchmarks, especially those stemming from previous work. How does the study address potential limitations or biases in the experimental design, and what steps were taken to ensure the reliability and validity of the results? Based on the findings, what are the potential avenues for further research on ribonucleotide and R-loop damage in plant organelle DNA and its role in maize development?

We added a conclusion section (lines 490-499, page 16) and included an idea for future work.

  1. Please check the reference section carefully and correct the inconsistency. Please update this section.

We rechecked the references and updated them according to the journal’s guidelines.

Round 2

Reviewer 3 Report

Reviewer 2:

I have reviewed the revised version manuscript entitled “Ribonucleotide and R-loop damage in plastid DNA and mitochondrial DNA during maize development”. The paper has been improved and can be accepted.

I have included my comments and I mentioned the English need to be polished but seems the authors ignore my comments:

It is better to do not to use the first-person's pronoun. Do not use "we, us, or our" throughout the paper.

Author Response

1.The author's response is in bold-I have included my comments and I mentioned the English need to be polished but seems the authors ignore my comments:

We incorporated some changes suggested by this reviewer, although in some cases this is a matter of preference in writing style.

2. It is better to do not to use the first-person's pronoun. Do not use "we, us, or our" throughout the paper.

Although we removed the first-person pronouns such as “we”, “our” in most places, we retained these pronouns in a few places. This is a matter of writing style preference.